# CREDIBLE, SEALED-BID, OPTIMAL REPEATED AUCTIONS WITH DIFFERENTIABLE ECONOMICS

## ABSTRACT

Online advertisement auctions happen billions of times per day. Bidders in auctions strategize to improve their own utility, subject to published auctions' rules. Yet, bidders may not know that an auction has been run as promised. A credible auction is one in which bidders can trust the auctioneer to run its allocation and pricing mechanisms as promised. It is known that, assuming no communication between bidders, no credible, sealed-bid, and incentive compatible (aka "truth-telling" or otherwise truthful-participation-incentivizing) mechanism can exist. In reality, bidders can certainly communicate, so what happens if we relax this (typically unrealistic) constraint?

In this work, we propose a framework incorporating cryptography to allow computationally-efficient, credible, revenue-maximizing (aka "optimal") auctions in a repeated auction setting. Our contribution is two-fold: first, we introduce a protocol for running repeated auctions with a verification scheme, and we show such a protocol can eliminate the auctioneer's incentive to deviate while costing negligible additional computation. Secondly, we provide a method for training optimal auctions under uncertain bidder participation profiles, which generalizes our protocol to a much wider class of auctions. Our empirical results show strong support for both the theory and competency of the proposed method.

## 1 INTRODUCTION

The problem of designing optimal, or revenue-maximizing, auctions bears significant theoretical and practical importance in economics: every Google search involves a sponsored search auction [1], webpage views involve real time auctions for ads, and online platforms like Ebay and Amazon have created markets ran by auctions. This problem is non-trivial: the auctioneer's revenue is dependent on the "best response" strategy of each bidder, which can each be dependent on each other. In his Nobel-prize-winning work, Myerson showed the $n$-bidder, 1-item optimal auction can be solved by essentially computing a virtual bid for each bidder, then maximizing welfare Myerson (1981); Daskalakis (2015). What about multi-item auctions? This has been shown to be no easy task, one clear reason for this difficulty is the size of the bundling space which grows exponentially. Additionally, an auctioneer may set reserve prices or draw lotteries to earn additional revenue. In essence, the optimal auction can be weird and "defying intuition" Daskalakis (2015).

Given no analytical solution have been found in designing the optimal multi-item auction, Daskalakis et al. (2014) have turned towards the complexity of this problem. They demonstrated that, under reasonable assumptions, finding the optimal multi-item auction is #P-hard. This has motivated the line of work called "differentiable economics" that focus on using machine learning to find desirable solutions to mechanism design problems Dütting et al. (2019), which includes auction design. Differentiable economics approaches consider an auction as a function that takes bids as inputs and returns what item is allocated to who and how much each bidder pays. This function is usually encoded as a neural network, which can be backpropagated on given a differentiable loss function. The loss function is parameterized by the revenue, incentive compatibility— which we will provide a definition and discuss in more detail in later sections— or other desirable properties

---

[1]A sponsored search auction is one where the website owner auctions different ad spots on the webpage when a certain keyword is searched.

of the auction Peri et al. (2021); Kuo et al. (2020). Although differentiable economics is a newly-emerged field, recent progress Dütting et al. (2019); Rahme et al. (2020; 2021a); Curry et al. (2021) show that it may be the most promising method for approximating optimal multi-item auctions.

Besides optimality, credibility of the auction is another major consideration. Consider a sealed-bid, one-item, second-price auction [2] being run between bidder 1 and bidder 2, whereas bidder 1 has valuation of $2 and bidder 2 has valuation of $3. Acting in their best strategy, bidder 1 and bidder 2 bid $2 and $3, and the auctioneer should then allocate the item to bidder 2 charging them $2. However, since the auction is sealed-bid, the auctioneer can tell bidder 2 that they won the auction and bider 1 bid $2.99, which would increase the auctioneer's revenue by $0.99. An auction is said to be credible if the auctioneer has no incentive not to stick to their proposed auction. The significance of auction credibility was brought to light when Google was called out for gaming their proposed second-price online ad auction Schiff (2022).

It is known that there exists no sealed-bid, incentive-compatible, and credible auction if communication between bidders is precluded Akbarpour & Li (2020). The authors of this work admit that modern cryptography along with bidder communication can potentially break the trilemma, but they consider the costs, in terms of computing resources and latency, of cryptographic constrictions too high.

In this paper, we propose an approach for running repeated auctions that greatly reduces the cost of a credible multi-item auction when verified either by revealing bids (i.e. we greatly reduce the number of bids that must be revealed) or using a cryptographic mechanism such as zero-knowledge proofs. First, we show that in a sequence of repeated auctions, we need not run the verification mechanism for every round. Instead, we can punish deviations with a penalty that when high enough, can prevent the auctioneer from being untruthful even when only a random set of auctions are audited. The repetition of auctions naturally brings up an issue regarding bidder participation which previous works in differentiable economics did not have to deal with, as it is unrealistic to assume that the same set of bidders participates from start to end in repeated auctions. We address this issue by proposing a model that takes account of bidder participation uncertainty, and we provide a method to extend previous works in differentiable economics to this model, which we support with experimental results.

## 2 RELATED WORK

**Neural networks**. RegretNet Dütting et al. (2019) was the first work to train incentive compatible auctions to maximize revenue using deep learning. RegretNet has two components: the allocation net and the payment net. Each network treats the corresponding part of the auction as a function, taking the bids from the bidders and outputting the allocation/payment. There are various version of other networks developed on the basis of RegretNet to cover specific needs. Peri et al. (2021) considers possible human preference in the allocation process. Kuo et al. (2020) focuses on improving the fairness of the auction mechanisms. There are also works focusing on improving the accuracy and efficiency of RegretNet. Rahme et al. (2020) proposed ALGNet as a more efficient version of RegretNet, which considers auction design as a auctioneer versus bidders adversarial model.

**Verification tools**. To prevent the auctioneer from deviating, we need some verification method that does not reveal additional information. Angel & Walfish (2013) proposed a cryptographic verification system called VEX that can be efficiently applied to second price auctions. In VEX, the auctioneer acts as the prover and the bidders act as the queriers. Under some given algorithm, the queriers can verify what the prover proposed is correct without information leakage in a considerable amount of time. More generally, Liu et al. (2021) has proposed zero-knowledge proof structures that can work for neural networks, and Mishra et al. (2020) has described a cryptography system which can also be applied to neural networks. With all that in mind, we can be confident that it is realistic to introduce verification tools during auction design.

There also exist concrete work on granting credibility in auction design for specific scenarios. Ferreira & Weinberg (2020) finds a credible and optimal auction for MHR valuations with commitment,

---

[2]In a second price auction, the auctioneer allocates the item to the highest bidder, and charges them the bid of the second highest bidder. The best strategy for any bidder in the second price auction is to bid exactly how much they value the item.

and Essaidi et al. (2022) extends that result to more general distributions of valuations. What separates this paper from theirs is that sealed-bid is not a consideration in these two papers, and our work focuses on a more general setting for neural network-encoded auctions.

# 3 BACKGROUND

## 3.1 AUCTION MODEL

We design a repeated auction with a set of $n$ bidders $N = \{1, 2, ..., n\}$ and $m$ items $M = \{1, 2, ..., m\}$ over some time horizon $T \in \mathbb{T}$, in which the bidders know how many bidders there are. This repeated auction does not partition one auction into several auctions that each sell a subset of the original items. Rather, each round of our repeated auction completes a multi-item auction, then the auctioneer restocks their goods and runs another stage auction. In our model, we assume each round of auction to be independent, and the bidders as memoryless agents that only try to maximize their utilities in the current round. So extensive-form game equilibria are not being considered, i.e. the bidders do not analyze how their actions in the current round can influence their utility in future rounds. We denote such repeated auctions $\mathbb{A}$.

During the $t$-th round, each bidder $i$ has a valuation function ${}^t v_i : 2^M \to \mathbb{R}_{\geq 0}$, where ${}^t v_i(S)$ denotes how much bidder $i$ values the set of items $S \subseteq M$ in round $t$. The valuation function ${}^t v_i$ is drawn independently from a distribution $F_i$ over possible valuation functions $V_i$, whereas $F_i$ is fixed in round $t$ and is public to both bidders and auctioneers. To provide a realistic simplification of the bidders' input space, we can assume the bidders have additive valuation, which means $v_i^t(S) = \sum_{s \in S} v_i^t(s)$. Therefore, we can use a matrix of size $n \times m$ to represent ${}^t v$. Upon receiving their valuation function at round $t$, bidder $i$ then reports their bids to the auctioneer. We will let ${}^t \theta_i \subseteq \mathbb{R}^m$ denote bidder $i$'s bids at round $t$, and let ${}^t \theta \subseteq \mathbb{R}^{n \times m}$ denote the full bid profile at round $t$. We will let the set $\Theta$ contain all possible bids, and we say bidder $i$ is truthful in round $t$ iff ${}^t \theta_i = {}^t v_i$.

Prior to the first round of the repeated auction $\mathbb{A}$, the auctioneer proposes a stage auction function as a tuple of an allocation function and a payment function $A = (a, p)$ whereas $a : R^{n \times m} \mapsto \{0, 1\}^{n \times m}$ and $p : R^{n \times m} \mapsto R^n$. We say $A = (a, p)$ is feasible if for any ${}^t \theta \in \Theta$, $\forall m \in M, \sum_{n \in N} a_{n,m} \leq 1$, in other words, no item is allocated more than once. In round $t$, once the auctioneer receives ${}^t \theta$, they invoke some feasible stage auction function ${}^t A = ({}^t a, {}^t p)$ to compute who gets what and how much they pay. We say the repeated auction $\mathbb{A}$ is truthful iff $\forall t \in T, {}^t A = A$.

The auctioneer's revenue in round $t$ is then ${}^t rev = \sum_{i=1}^n {}^t a_i({}^t \theta) \cdot {}^t p_i({}^t \theta)$, whereas ${}^t a_i(x), {}^t p_i(x)$ represent the $i$-th row of ${}^t a(x), {}^t p(x)$. Traditional auction design studies ask the auctioneer to publish the mechanism function prior to the auction, and follow it strictly. In the work by Akbarpour & Li (2020), they considered the auctioneer as a utility-maximizing agent as well, which opened up the doors to the study of auction credibility. We take a similar approach in our model. We first ask the auctioneer to publish the mechanism, however, the auctioneer is free to deviate from this plan in strategic ways, which means it may be the case that the auctioneer can obtain higher revenue by running some $A' \neq A$ in round $t \in T$, thus we define the regret of the auctioneer in round $t$ as

$$ {}^t \text{aRgt} = \max_{{}^t A}[\text{rev}({}^t A, {}^t \theta)] - \text{rev}(A, {}^t \theta) $$

such that ${}^t A$ is feasible. Similarly, the utility of bidder $i$ is $u_i({}^t A, {}^t \theta, {}^t v) = {}^t a_i({}^t \theta) \cdot [{}^t v_i - {}^t p_i({}^t \theta)]$. It may be the case that a bidder can obtain higher utility by misreporting, we formalize this by defining the regret of a bidder as

$$ {}^t \text{uRgt}_i = \max_{{}^t \theta_i}[u_i({}^t A, {}^t \theta, {}^t v)] - u_i({}^t A, {}^t v, {}^t v) $$

whereas ${}^t \theta_{-i} = {}^t v_{-i}$. This is equivalent to searching for bidder i's optimal misreport assuming all other bidders are truthful in round $t$.

### 3.1.1 STATIONARY PARTICIPATION

Traditional auction design and differentiable economics studies usually define a stage auction for a fixed set bidders and a fixed set of items. Our model won't require each bidder to show up to every round of $\hat{A}$. Rather, we will use $g$ to denote a **participation profile**, which is a binary string of

length $n$ that indicates which bidders showed up to the auction. The set $G \subseteq \{0, 1\}^n$ will contain all possible $g_i$s and the set $Q$ will contain $q_i$s that entail the probability of $g_i$ at any round $t \in [T]$. This implies the probability distribution of participation profiles is stationary, and we call this the **stationary participation model**. We will use **fixed-bidder model** to denote the special case where $|G| = 1, G = \{< 1, 1, 1, ..., 1 >\}$

### 3.2 CRYPTOGRAPHIC BACKGROUND

**Commitments** A commitment scheme is a cryptographic protocol that allows a user to commit to data by publishing the commitment without revealing the actual data. Given the data and a commitment, anyone can verify that the data has not been changed since the commitment was published. Commitment schemes are said to be binding—once committed, data cannot be changed even by the original owner—and hiding—the commitment on it's own does not reveal the data. A simple efficient commitment scheme consists of picking a 128 bit random number and hashing it[3] together with the data to be committed to.

**Zero-Knowledge Proof** A zero-knowledge proof allows a prover to convince one or more verifiers that some statement hold without revealing how or why. Goldreich et al. (1986) have shown that there exists a zero-knowledge proof for any NP-relation thus there exists a zero-knowledge proof for the correctness of auctions. More concretely, the last decade as seen marked advances practical zero-knowledge proof systems to the point where they can efficiently handle matrix multiplications and even neural networks. Liu et al. (2021) build a non-interactive zero-knowledge proof for predication in neural networks: given a model, the zero-knowledge proof shows that its output is correct for given inputs.

## 4 PROTOCOL

We define, in Algorithm 1, the protocol that we propose. For the sake of comparison, we also define (Appendix: A.1) a default auction protocol if companies like Google were to adopt recent works in differentiable economics like Dütting et al. (2019); Rahme et al. (2020); Peri et al. (2021) into their sponsored search auctions.

---

**Algorithm 1** Proposed Repeated Auction Protocol

Bidder valuation distributions made public
Auctioneer proposes $A = (a, p)$
Initialize $T \in \mathbb{N}$
Initialize logs $L_\theta, L_a, L_p \in R^{T \times n \times m}$
Initialize penalty $\in (\{0\} \cup \mathbb{R}^+)$
$t \leftarrow 1$
**while** $t \leq T$ **do**
    $L_\theta[t, :, :] \leftarrow^t \theta$
    auctioneer solicit $^t\theta$ from bidders
    auctioneer runs auction with $^t\theta$ to obtain $^t a(^t\theta), ^t p(^t\theta)$
    $L_a[t, :, :] \leftarrow ^t a(^t\theta)$
    $L_p[t, :, :] \leftarrow ^t p(^t\theta)$
    $t \leftarrow t + 1$
**end while**
bidders randomly select $r \in \{1, ..., T\}$
$s \leftarrow \text{ver}(A, L_\theta[r, ::], L_a[r, ::], L_p[r, ::])$
**if** $s = 0$ **then**
    penalize auctioneer by penalty
**end if**

---

whereas the verification function $ver$ is defined below.

---
[3]This holds for hash functions like SHA3.

**Definition 4.1.** *The verification function* $\text{ver}(A, {}^t\theta, {}^ta', {}^tp')$ *takes in a stage auction* $A = (a, p)$, *the bids of the bidders* ${}^t\theta$ *in round t, along with a hypothesis allocation matrix* ${}^ta' \in \mathbb{R}^{n \times m}$ *and a hypothesis payment matrix* ${}^tp' \in \mathbb{R}^{n \times m}$. *It returns 1 if*

$$ {}^ta' = a({}^t\theta), {}^tp' = p({}^t\theta), $$

*otherwise returns 0.*

The log in algorithm 1 can be implemented with commitments and access to a broadcast channel which ensures all parties see the committed bids when they are announced. Upon request, the log provides the bids and results of the auction in a specific round, which can be audited. We have abstracted the audit process as the verification function $ver$ because it can take on various forms. The most straightforward way to accomplish $ver$ is to bring in a trusted third party, possibly at a cost that the bidders and auctioneer pay together. This third party can simply take the bids in round $r$ and run it with the proposed auction function. Other than incorporating a third-party, the bidders can ask the auctioneer to publish the result of the auction in round $r$, which the bidders can then verify with their commitments in round $r$. The downside of this approach is that the allocation and payment of a randomly selected round will be revealed, so a bidder can possibly learn the bidding strategy of another. To address this issue, the bidders can construct a zk-SNARK (Zero-Knowledge Succinct Non-Interactive Argument of Knowledge) Liu et al. (2021) out of the auction function, which would prevent the leakage of private information during verification.

Next, we will show that the verification scheme along with the penalty are sufficient to make the auction credible while preserving desirable properties of the stage auction. From now on, when we write $\mathbb{A}(A)$, we refer to Algorithm 1 implemented with $A$ as its proposed stage auction.

### 4.1 Obtaining Credibility

Given the proposed stage auction function $A$, let $\text{aRgt}^*$ be an upper bound on ${}^t\text{aRgt}$ over all possible reported bids in $A$. This bound certainly exists if the bidders' valuation distributions are bounded, even if they are not bounded in the model, it would be reasonable to assume they are. In practice, this bound can be found by summing the highest market price of every item. Then the additional revenue that the auctioneer makes by being untruthful in $w$ out of $T$ rounds is bounded by $w \cdot \text{aRgt}^*$. Since in algorithm 1, the bidders are selecting a round to verify with a uniform distribution, the probability that the auctioneer is not caught deviating is $1 - \frac{w}{T}$. So with the penalty considered, the expected additional revenue in each untruthful round is

$$ \frac{1}{\omega}[(1 - \frac{w}{T})w \cdot \text{aRgt}^* - \frac{w}{T}\text{penalty}] $$

$$ = \text{aRgt}^* - \frac{w \cdot \text{aRgt}^*}{T} - \frac{\text{penalty}}{T} \leq \text{aRgt}^* - \frac{\text{penalty}}{T}. $$

Notice that the above inequality is independent of $w$, and when penalty $> T \cdot \text{aRgt}^*$, this expression is negative. So if we can estimate $\text{aRgt}^*$, we can set a penalty so that the auctioneer makes negative expected additional revenue per untruthful round. This should prevent the auctioneer from deviating in any round. We include a strategy for the auctioneer to make additional revenue by being untruthful when the penalty is not high enough in appendix A.1.

We have identified an approach to obtain truthful auctioneers using a verification scheme and penalty. We now turn to the problem of maximizing the auction's revenue, which is concerned with the bidders' behaviors. Since we have shown credibility can be obtained independent of bidders' behaviors, we can assume that the auctioneer will be truthful to their proposed auction from now on.

### 4.2 Bidder Behavior

In practice, it is difficult to predict behaviors of bidders under a certain mechanism. However, using the concept of equilibrium and incentive compatibility, we can infer some behaviors of the bidders assuming rationality and perfect information. We provide the definition of incentive compatibility in two solution concepts below.

**Definition 4.2.** *A stage auction $A$ is **Bayesian Nash Incentive Compatible (BNIC)** if there is a Bayesian Nash equilibrium where the bidders report their true valuation. A repeated auction $\mathbb{A}$ is **BNIC** if given every other bidder choose to be truthful in every round, then bidder $i$ achieves their optimal utility by bidding truthfully in every round.*

**Definition 4.3.** *A stage auction $A$ is **Dominant Strategy Incentive Compatible (DSIC)** if being truthful weakly dominates every other strategy regardless of other bidders' strategy. A repeated auction $\mathbb{A}$ is **DSIC** if no matter what every bidder does, bidder $i$ achieves their optimal utility by bidding truthfully in every round.*

The revelation principle states that for any auction mechanism $A = (a, p)$ such that there exists a bidding equilibrium under $A$, there exists a mechanism $A' = (a', p')$ such that $A'$ is incentive compatible (BNIC/DSIC depending on solution concept of the equilibrium) and achieves the same payoff profile as $A$ in expectation. The adoption of incentive compatibility is commonplace in mechanism design studies due to this fact, which allows us to search in the smaller space of incentive compatible auctions when looking to maximize revenue.

It's not hard to see that in both stage auctions and repeated auctions, $DSIC$ implies $BNIC$. In fact, in the fixed-bidder model, the repeated auction $\mathbb{A}(A)$ will inherit incentive compatibility properties of the stage auction $A$.

**Lemma 4.4.** *In the **fixed bidder model**, $\mathbb{A}(A)$ is DSIC iff $A$ is DSIC, similarily, $\mathbb{A}(A)$ is BNIC iff $A$ is BNIC.*

**Corollary 4.4.1.** *In the **fixed bidder model**, if $A$ is revenue-maximizing, DSIC/BNIC. Then if we set penalty $> T \cdot \mathrm{aRgt}^*$ in $\hat{A}$, $\mathbb{A}(A)$ is credible, DSIC/BNIC, and revenue maximizing.*

The proofs of the above theorems are in appendix A.4.1. This implies to find the revenue-maximizing repeated auction $\mathbb{A}(A)$ in the fixed bidder model, we can just use the machine learning-based techniques proposed by Dütting et al. (2019) to optimize $A$.

Now we transition to the stationary participation model, where the same set of items are being auctioned each round, but the bidders may change according to some stationary probability distribution. In this model, despite the auctioneer's uncertainty about which bidder will participate in any of the future rounds, they can still observe which bidder participates in the current round, as we can emulate a non-participating bidder by assuming their valuation for each item is 0. Therefore, the auctioneer can design a stage auction mechanism that depends on which bidder participates.

Let $G$ contain all participation profiles that happen with non-zero probability, and let ${}^t g \in G$ be the participation profile in the $t$-th round. The auctioneer will use an **aggregated auction** $\hat{A}$ as their proposed stage auction whereas $\hat{A}$ is defined by a mapping $d({}^t \theta) : \Theta \mapsto G$ and a set $agg(\hat{A})$ that contains an auction $A_i$ for each $g_i \in G$, which means $|agg(\hat{A})| = |G| \leq 2^n$. Specifically, the aggregated auction $\hat{A}$ is defined by the following piecewise function:

$$\hat{A}({}^t \theta) = \begin{cases} A_1({}^t \theta) & \text{if } d({}^t \theta) = g_1 \\ A_2({}^t \theta) & \text{if } d({}^t \theta) = g_2 \\ ... & ... \\ A_{|G|}({}^t \theta) & \text{if } d({}^t \theta) = g_{|G|} \end{cases}$$

whereas the mapping $d$ can be accomplished by rounding all non-zero entries in ${}^t \theta$ to 1, and then find the maximum bid of each bidder. This will result in a binary vector that must correspond to its matching $g_i \in G$. It's not hard to see that in the stationary participation model, any stage auction function $A$ can be written in the form of an aggregated auction $\hat{A}$ which consists of a mapping from bids to a set of auctions. Therefore, we will say that the auctioneer selects an aggregated auction as their proposed auction function in the stationary participation model.

**Lemma 4.5.** *In the **stationary participation model**, if for any $A_i \in agg(\hat{A})$, $A_i$ is DSIC/BNIC, then $\mathbb{A}(\hat{A})$ is DSIC/BNIC.*

**Theorem 4.6.** *In the **stationary participation model**, the repeated auction $\mathbb{A}(\hat{A})$ is revenue-maximizing iff $A_i$ is revenue-maximizing for any $A_i \in agg(\hat{A})$.*

The proofs of the above statements are in appendix A.4.1. From now on, we use $\hat{A}^*$ to refer to a revenue-maximizing instance of $\hat{A}$. Theorem 4.6 informs us that to find $\hat{A}^*$, it suffices to find a set

of auctions where each auction corresponds to a participation profile and is revenue-maximizing. we now discuss our approach to this task.

## 5    ESTIMATING $\hat{A}^*$

We expect the size of $agg(\hat{A}^*)$ to grow quickly with respect to the number of bidders. In particular, if the bidders' participation probabilities are independent from each other, the size of $agg(\hat{A}^*)$ will grow exponentially. Considering the known complexity of finding a single revenue-maximizing multi-item auction, we can infer that the task of obtaining $\hat{A}^*$ is daunting. This complexity is somewhat relieved by recent works in differentiable economics, as if we can settle with an approximately optimal auction for each participation profile, each $A_i^* \in agg(\hat{A}*)$ can be approximated with machine learning. However, we would still need to perform auction training $|G|$ number of times, which can be exponential. This section presents an approach to circumvent this complexity, and its performance is experimentally evaluated in the subsequent section.

### 5.1    TWEAKED DATASET

Our insight for estimating $\hat{A}^*$ is to generate a dataset according to $G$ (participation profiles) and $Q$ (probabilities of profiles) under the stationary participation model, we call this dataset the "tweaked dataset". Then we try to train an auction that performs well in the tweaked dataset in expectation. To put simply, we are estimating every element of $agg(\hat{A}^*)$ at the same time. The tweaked dataset will contain a size $K$ number of frames whereas each frame will be a matrix of size $n \times m$ that contains the valuations of each bidder for each object. To obtain each frame, we first sample a participation profile $g \in G \subseteq \{0,1\}^n$ according to $Q$, then sample the untweaked valuations $v \in \mathbb{R}^{n \times m}$, then element-wise multiply the two after broadcasting $g$ across the items' dimension.

### 5.2    ARCHITECTURE

We adopt the additive neural network architecture from Rahme et al. (2020), which consists of a multi-layer perceptron (MLP) allocation and payment network. Similar styles of mechanism neural network architectures are used in Duetting et al. (2019); Duan et al. (2022); Ivanov et al. (2022). Since the auctioneer can choose not to allocate an item, and the optimal auction can take on the form of a lottery, the allocation network is implemented as two networks. The first one ($f_1 : \mathbb{R}^{n \times m} \mapsto [0,1]^m$) computes the probability that the auctioneer allocates each item; the second one ($f_2 : \mathbb{R}^{n \times m} \mapsto [0,1]^{n \times m}$) computes the probability that an item will be allocated to each bidder if the auctioneer allocates that item. In Rahme et al. (2020) and our implementation, $f_1(\theta) = \sigma(\text{MLP}(B))$, $f_2(\theta) = \text{softmax}(\text{MLP}(B))$ (allowing for ghost bidders/items representing "no allocation"). The final allocation is then obtained with $a_{i,j} = [f_1(B)]_{i,j} \cdot [f_2(B)]_{i,j}$. The payment function is computed as $p = \sigma(\text{MLP}(\theta))$, a ratio of the bidder's bid that they shall pay.

### 5.3    REGRET ESTIMATION

To compute the bidder's regret, we follow the approach proposed by Duetting et al. (2019) and use a misreport optimization loop to estimate the optimal "untruthful bid" of each bidder assuming other bidders are truthful. The misreport function is a MLP whose width and depth will be specified. The output of the misreport network is a $n \times m$ matrix of ratios between 0 and 1, which when element-wise multiplied by the valuations returns the matrix of misreports. This allows the individual rationality constraint to be built into the network. When testing the misreport module, we noticed that there does not seem to be a general optimal misreport network: one misreport network can perform excellently for one auction but horribly for another. Therefore (similar to the choice of Rahme et al. (2021b)), we reinitialize the weights of the misreport network at each iteration of the auction training loop. We also found that the efficiency of the regret estimation step is greatly improved if we allow early stopping of the misreport, which means stopping the misreport optimization loop once the regret stops increasing for a certain number of rounds. Hyperparameters for early stopping will be specified in section 6.

## 5.4 TRAINING

The training loop consists of three steps: 1) computing the auctioneer's revenue and bidders' regret 2) computing loss and gradient 3) backpropagation. A nice property of thee loss function would be providing a comparison between two auctions with different revenue and regret. This is accomplished using Proposition 1 from Rahme et al. (2020), attributed to Balcan et al. (2005) and Nisan.

**Lemma 5.1.** *(Rahme et al. (2020)) Let* rev, rgt *denote the expected revenue and regret of some auction, then there must exist some other auction that achieves zero BNIC regret and revenue of* $(\sqrt{\text{rev}} - \sqrt{\text{rgt}})^2$ *under the same setting.*

The lemma above can be applied to DSIC auctions in 1 bidder, $m$ item auctions. Whether it also holds for general $n$ bidder, $m$ item DSIC auctions is still an open problem Rahme et al. (2020). Regardless, for any auction with non-zero regret, it provides a lower bound for the revenue of its zero-regret counterpart in BNIC solution concept. Nevertheless, Rahme et al. (2020) argues convincingly that it is a reasonable single metric to compare auctions which we can estimate the competency of any non-zero regret learned auction. We will adopt the above lemma as the loss function, which is stated below.

$$loss = -(\sqrt{\text{rev}'} - \sqrt{\text{rgt}'}) - \text{rgt}'.$$

(The $-rgt'$ term is for making the model slightly inclined towards auctions with low regret.)

## 6 EXPERIMENTS

We perform experiments to test the capability of our proposed training procedure to recover $\hat{A}^*$. We first choose the n = 2, m = 2, bidder valuation uniformly distributed between $[0, 1]$ case, because it is a classic test case for automated mechanism design Sandholm & Likhodedov (2015); Likhodedov & Sandholm (2005), widely used by related papers in differentiable economics Duetting et al. (2019); Duan et al. (2022); Ivanov et al. (2022); Rahme et al. (2021a;b); Curry et al. (2020) and allows an in-depth evaluation of the results. In particular, we pick three scenarios to test on: 1. bidder 1 participates with 0.2 probability, bidder 2 participates with 0.8 probability, 2. bidder 1 participates with 0.5 probability, bidder 2 participates with 0.7 probability, 3. bidder 1 participates with 0.7 probability, bidder 2 participates with 0.9 probability. Since the participation probabilities of bidders are all independent, the participation profiles for each scenario are the same, namely $g_1 = <1, 1>$, $g_2 = <1, 0>$, $g_3 = <0, 1>$, $g_4 = <0, 0>$. However, the probabilities of each profile are different across scenarios and are specified below.

- Scenario 1: $q_1 = 0.16, q_2 = 0.04, q_3 = 0.64, q_4 = 0.16$
- Scenario 2: $q_1 = 0.35, q_2 = 0.15, q_3 = 0.35, q_4 = 0.15$
- Scenario 3: $q_1 = 0.63, q_2 = 0.07, q_3 = 0.27, q_4 = 0.03$

All of the scenarios above are trained on their corresponding tweaked dataset with $K = 20,000$, and the depth, width of the allocation MLP, payment MLP, and misreport MLP are all set to 7 and 100. The learning rate of the misreport optimizer is set to $1 \times 10^{-5}$ and it is looped for 300 times with early stopping point set to 100. The auction function optimization step is looped for 300 times with learning rate of $5 \times 10^{-4}$ for the first 100 iterations, $5 \times 10^{-5}$ for the second 100 iterations, and $5 \times 10^{-6}$ for the last 100 iterations. After training, we evaluate the learned auctions on a separate test set for both the aggregated auction and the individual auction performance. We first report the results on the three scenarios above when trained on tweaked datasets. Note that each scenario is trained with a designated neural network, and tested on that network.

Table 1: Performance of three learned auctions in their corresponding scenario after training on the tweaked dataset

| Scenario | $\text{rev}'_{agg}$ | $\text{rev}^*_{agg}$ | $\text{rgt}'_{agg}$ | $\left(\sqrt{\text{rev}'_{agg}} - \sqrt{\text{rgt}'_{agg}}\right)^2$ |
|---|---|---|---|---|
| 1 | 0.470 | $\approx 0.513$ | $5 \times 10^{-3}$ | 0.375 |
| 2 | 0.580 | $\approx 0.579$ | $9 \times 10^{-3}$ | 0.444 |
| 3 | 0.765 | $\approx 0.735$ | 0.013 | 0.577 |

For comparison, we also train an auction function on an untweaked dataset, and test it on the test sets of the three scenarios above.

Table 2: Performance of a neural network trained on untweaked dataset tested on the three scenarios

| Scenario | $\text{rev}'_{agg}$ | $\text{rev}^*_{agg}$ | $\text{rgt}'_{agg}$ | $\left(\sqrt{\text{rev}'_{agg}} - \sqrt{\text{rgt}'_{agg}}\right)^2$ |
|---|---|---|---|---|
| untweaked | 0.854 | $\approx 0.87$ | $3 \times 10^{-4}$ | 0.822 |
| 1 | 0.279 | $\approx 0.513$ | $9 \times 10^{-4}$ | 0.247 |
| 2 | 0.422 | $\approx 0.579$ | $3 \times 10^{-3}$ | 0.352 |
| 3 | 0.615 | $\approx 0.735$ | $4 \times 10^{-3}$ | 0.514 |

The comparison between table 1 and table 2 show that the networks trained on the tweaked dataset perform markedly better than the benchmark. In theory, $\hat{A}^*$, the optimal auction under stationary participation profile is simply a "weighted sum" of its individual auctions. So we should expect the learned auctions from each of the three scenarios to perform decently in each individual participation profile. We evaluated this by testing the three neural networks trained on the tweaked datasets on the $2 \times 2$ and $1 \times 2$ scenarios additionally.

Table 3: Benchmark performance of learned aggregated auction on individual auctions

| NN | $g_1 = <1,1>$ | | | $g_2 = <1,0>$ | | | $g_3 = <0,1>$ | | |
|---|---|---|---|---|---|---|---|---|---|
| | $\text{rev}'_1$ | $\text{rev}^*_1$ | $\text{rgt}'_1$ | $\text{rev}'_2$ | $\text{rev}^*_2$ | $\text{rgt}'_2$ | $\text{rev}'_3$ | $\text{rev}^*_3$ | $\text{rgt}'_3$ |
| 1 | 0.688 | $\approx 0.87$ | $8 \times 10^{-4}$ | 0.031 | 0.55 | $3 \times 10^{-5}$ | 0.563 | 0.55 | $7 \times 10^{-3}$ |
| 2 | 0.886 | $\approx 0.87$ | 0.019 | 0.503 | 0.55 | $5 \times 10^{-3}$ | 0.563 | 0.55 | $5 \times 10^{-3}$ |
| 3 | 0.927 | $\approx 0.87$ | 0.017 | 0.473 | 0.55 | $2 \times 10^{-4}$ | 0.535 | 0.55 | $2 \times 10^{-3}$ |

We see that the learned auctions perform relatively well on the individual auctions: achieving a revenue close to the optimal in most cases while maintaining low regret. We see a trend that the performance on the individual auctions is dependent on the probability of the participation profile associated with that auction. For example, in scenario 1, the participation profile $g_2 = <1,0>$ happens with a low probability of $0.04$, so for the first neural network, if it performs badly in $g_2$ it won't harm the aggregated auction performance as much as if it performs badly in $g_3 = <1,1>$, which happens with probability 0.64. Therefore, although $g_2$ and $g_3$ are in theory the same auction, the first neural network performs better in $g_3$. This trend can also be found in the other two scenarios. A larger scale experiment with 3 bidders and 10 items is included in Appendix A.5.

## 7 CONCLUSION

In this paper, we demonstrate how to run credible, incentive compatible, privacy-preserving and revenue maximizing auctions in settings where auctions take place with high frequency. Our work is inspired by the impossibility theorem proposed by Akbarpour & Li (2020), where a trilemma is established between credibility, incentive compatibility, and privacy-preserving in stage auctions assuming no communication between bidders. Because cryptographic protocols are efficient these days, we relax the assumption of no bidder communication, and show that by implementing a verification scheme in a repeated auction, we can obtain credibility while maintaining incentive compatibility and bidders' privacy. We also propose a stationary bidder participation model, which to our knowledge is the first in the differentiable economics community. We provide a method for training revenue-maximizing auctions in the stationary participation model, whose theory and efficacy is tested with two experiments. We note that our method for training revenue-maximizing auctions in the stationary participation model can not only be applied in our repeated auctions protocol, but also any stage auction where participation of bidders is uncertain.

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

# A  Appendix

## A.1  Benchmark Algorithm

The following algorithm provides a default way to implement auctions learned with differentiable economics techniques Dütting et al. (2019) in a high-frequency auction market such as online ad auctions.

---

**Algorithm 2** Default Repeated Auction Protocol

---

    Bidder valuation distributions made public
    Auctioneer proposes $A = (a, p)$
    Initialize $T \in \mathbb{N}$
    $t \leftarrow 1$
    **while** $t \not\succ T$ **do**
        auctioneer solicit $^t\theta$ from bidders
        auctioneer runs auction with $^t\theta$ to obtain $^ta(^t\theta), {}^tp(^t\theta)$
        $t \leftarrow t + 1$
    **end while**

---

Note that the running time for our proposed auction protocol described in algorithm 1 and the default protocol algorithm 2 are both linear with respect to $T$ assuming the auction, queries of the log, and the verification scheme $ver$ take constant time.

## A.2  Sample Untruthful Strategy for Auctioneer

We now illustrate a strategy for the auctioneer when the penalty is not set high enough in Algorithm 1. Recall that this means penalty $\leq T \cdot \text{aRgt}^*$. Since the valuation distribution is fixed, then $^t\theta$ shall be sampled from the set of all possible bids $\Theta \subseteq \mathbb{R}^{n \times m}$. Suppose there exists a set of misreports

$$\Theta' = \{\theta' \mid \theta' \in \Theta, aRgt(A', \theta') \geq \text{penalty}/T \text{ for some feasible } A', \text{Prob}(\theta') > 0\},$$

then if the auctioneer deviates whenever $^t\theta \in \Theta'$ and be truthful otherwise, they obtain equal or higher revenue compared to the strategy of always being truthful.

## A.3  Regret and Revenue of Individual Auctions

**Corollary A.0.1** (Corollary of Lemma 5.1 by Rahme et al. (2020)). *Let $rev^*$ be the optimal revenue for an additive auction with $n$ bidders and $m$ items. Then under the BNIC solution concept, for any other auction $A'$ that achieves expected revenue $rev'$ and expected mean regret $rgt'$, the following inequality must hold,*

$$rev^* \geq (\sqrt{rev'} - \sqrt{rgt'})^2.$$

The above corollary allows us to find an upper bound for the revenue of an auction given its regret and the optimal revenue, which is useful in estimating the revenue of an individual auction within an aggregated auction. We will discuss this below.

In section 5.3 and 5.4, we have transformed estimating $\hat{A}^*$ into learning a revenue-maximizing auction in a setting where bidders have a "tweaked" valuation distribution. By definition of the aggregated auction $\hat{A}$, if we obtain $\hat{A}'$ as a decent estimate of $\hat{A}^*$, we should expect at least some $A_i \in agg(\hat{A}')$ to be near revenue-maximizing as well. For example, if we are given $n = 3, m = 2$, all bidders have valuation uniformly distributed in $[0, 1]$, and $G = \{< 1, 1, 1 >, < 1, 0, 0 >\}$, and we learn a competent auction $\hat{A}'$ for this stationary participation model, then we should expect $\hat{A}'$ to perform well in the fixed-bidder 1 bidder 2 item auction as well. Thankfully, this is something we can check because analytical results are known for special cases of combinatorial auction including a 1 bidder 2 item case Manelli & Vincent (2006).

We now let $\hat{A}'$ denote the learned aggregated auction, and let $\mathrm{rev}'_i$, $\mathrm{rgt}'_i$ denote the expected revenue and regret of $A'_i \in agg(\hat{A}')$ which corresponds to the participation profile $g_i$ that happens with probability $q_i$. We will let $rev_i^*$ denote the revenue of the optimal auction for participation profile $g_i$. Suppose the aggregated auction $\hat{A}'$ achieves revenue $\mathrm{rev}'_{agg}$ and regret $\mathrm{rgt}'_{agg}$, then we know the following about the performance of individual auctions $A'_i$s.

**Lemma A.1.** *The regret of the individual auctions can be bounded in the following way* $\mathrm{rgt}_i \leq \frac{\mathrm{rgt}_{super}}{q_i}$.

*Proof of Lemma A.1.* The aggregated regret $rgt_{agg}$ is a weighted sum of the regret of each of its individual auctions $rgt_{agg} = \sum_{i \in |G|} q_i \cdot rgt_i$. □

**Lemma A.2.** *The revenue of the individual auctions can be upper bounded in the following way*

$$rev_i \leq \left( \sqrt{rev_i^*} + \sqrt{\frac{\mathrm{rgt}_{agg}}{q_i}} \right)^2,$$

*whereas* $rev^*$ *is the optimal revenue.*

*Proof of Lemma A.2.* From Corollary 5.1.1, we know $\sqrt{rev'} \leq \sqrt{rev^*} + \sqrt{rgt'}$. Plugging in $rgt' \leq \sqrt{\frac{rgt_{super}}{q_i}}$, we obtain the claim. □

Thus if we know the optimal revenue of the individual auctions in $\hat{A}$, and we know the revenue and regret of the learned auction $\hat{A}'$, we can find upper bounds of the revenue and regret of each individual learned auction. Since the aggregated revenue is a sum of the revenue of each individual auctions weighted by their corresponding probability, we can also obtain a lower bound for the revenue of the $i$-th auction by subtracting away the upper bound of every other auction.

**Theorem A.3.** *The revenue of* $A'_i \in agg(\hat{A}'))$ *is guaranteed in the following way*

$$rev'_i - rev_i^* \geq \frac{rev'_{agg} - rev_{agg}^*}{q_i} - \frac{|G| - 1}{q_i} \cdot rgt'_{agg} - 2 \frac{\sqrt{rgt'_{agg}}}{q_i} \sum_{\substack{j=1 \\ j \neq i}}^{|G|} \left( \sqrt{q_j} \sqrt{rev_j^*} \right).$$

*Proof of Theorem A.3.* We can plug lemma A.2 into

$$rev'_{agg} = \sum_{i=1}^{|G|} q_i rev'_i$$

to obtain

$$q_i \cdot rev'_i \geq rev'_{agg} - \sum_{\substack{j=1 \\ j \neq i}}^{|G|} q_j \left( \sqrt{rev_j^*} + \sqrt{\frac{rgt'_{agg}}{q_j}} \right)^2$$

expanding the right hand side leaves

$$q_i \cdot rev'_i \geq rev'_{agg} - \sum_{\substack{j=1 \\ j \neq i}}^{|G|} q_j \left( rev_j^* + \frac{rgt'_{agg}}{q_j} + 2\sqrt{rgt_j^*} \sqrt{\frac{rgt'_{agg}}{q_j}} \right)$$

$$q_i \cdot rev'_i \geq rev'_{agg} - (|G| - 1) \cdot rgt'_{agg} - \sum_{\substack{j=1 \\ j \neq i}}^{|G|} q_j \cdot rev_j^* - 2\sqrt{rgt'_{agg}} \sum_{\substack{j=1 \\ j \neq i}}^{|G|} \left( \sqrt{q_j} \sqrt{rev_j^*} \right)$$

notice that

$$-(q_i \cdot rev_i^* + \sum_{\substack{j=1 \\ j \neq i}}^{|G|} q_j \cdot rev_j^*) = -\sum_{j=1}^{|G|} q_j rev_j^* = -rev^*,$$

$$-\sum_{\substack{j=1 \\ j \neq i}}^{|G|} q_j \cdot rev_j^* = -rev^* + q_i \cdot rev_i^*,$$

plugging it in leaves

$$q_i \cdot rev_i' \geq rev_{agg}' - (|G| - 1) \cdot rgt_{agg}' - rev_{agg}^* + q_i \cdot rev_i^* - 2\sqrt{rgt_{agg}' \sum_{\substack{j=1 \\ j \neq i}}^{|G|} \left( \sqrt{q_j} \sqrt{rev_j^*} \right)},$$

$$q_i \cdot rev_i' - q_i \cdot rev_i^* \geq (rev_{agg}' - rev_{agg}^*) - (|G| - 1) \cdot rgt_{agg}' - 2\sqrt{rgt_{agg}' \sum_{\substack{j=1 \\ j \neq i}}^{|G|} \left( \sqrt{q_j} \sqrt{rev_j^*} \right)},$$

$$rev_i' - rev_i^* \geq \frac{rev_{agg}' - rev_{agg}^*}{q_i} - \frac{|G| - 1}{q_i} \cdot rgt_{agg}' - 2\frac{\sqrt{rgt_{agg}'}}{q_i} \sum_{\substack{j=1 \\ j \neq i}}^{|G|} \left( \sqrt{q_j} \sqrt{rev_j^*} \right).$$

$\square$

Note that theorem A.3 is a quite conservative bound, because lemma A.1 is a conservative bound for $rgt_i'$, and theorem A.3 repeatedly applies it $|G| - 1$ many times. Therefore as $|G|$ increases, the bound in theorem A.3 will become loose pretty quickly. We also provide an alternative lower bound for $rev_{agg}'$ that is tighter than theorem A.3, the tradeoff is that this tighter bound contains a maximization problem.

**Proposition A.4.** *The revenue of $A_i' \in agg(\hat{A}')$ is guaranteed in the following way*

$$q_i \cdot rev_i' \geq rev_{agg}' - \max_u \left[ \sum_{\substack{j=1 \\ j \neq i}}^{|G|} q_j \cdot \left( \sqrt{rev_i^*} + \sqrt{u_j \cdot rgt_i'} \right)^2 \right]$$

*under the constraint that $u \in [0,1]^{|G|}$ and $\sum_{i=1}^{|G|} u_i = 1$.*

The maximization problem inside this bound can make it seem complex, in fact, the objective function of the maximization problem is quite straightforward, thus the bound can be computed efficiently as well.

## A.4 PROOFS

### A.4.1 PROOFS IN SECTION 4

*Proof of Lemma 4.4.* To see that $\mathbb{A}(A)$ is BNIC implies $A$ is BNIC and $\mathbb{A}(A)$ is DSIC implies $A$ is DSIC, observe that $A$ is a special case of $\mathbb{A}(A)$ where $T = 1$. For the other direction, suppose $A$ is BNIC and every bidder but $i$ is truthful in each of the $t$ rounds, then always being truthful weakly-dominates every other possible strategy for bidder $i$. The same argument goes for DSIC. $\square$

*Proof of Lemma 4.5.* Let $u_{i,j}$ be the expected utility of bidder $i$ by bidding truthfully in $A_j \in agg(\hat{A})$ regardless of other bidders' strategy. By the assumption that every $A_j \in agg(\hat{A})$ is DSIC, $u_{i,j}$ must be the highest utility that bidder $i$ can obtain in $A_j$. Now suppose $\mathbb{A}(\hat{A})$ reaches round $t$, which means the auctioneer will run some $A_j \in agg(\hat{A})$ that corresponds to $^t g$. Despite the bidder may not know what $A_i$ is, the maximum expected utility for bidder $i$ in this round is $u_{i,j}$, which is achieved by bidding truthfully. The proof for BNIC follows a similar argument where we relax the assumption to expect other bidders to be truthful. □

*Proof of Theorem 4.6.* The expected revenue per round of $\mathbb{A}(\hat{A})$ can be computed as

$$\sum_{g_i \in G} q_i \cdot (\text{expected revenue of } \hat{A} \text{ at } g_i) = \sum_{i=1}^{|G|} q_i \cdot (\text{expected revenue of } A_i).$$

It's then clear that the expected revenue per round of $\mathbb{A}(\hat{A})$ can be improved iff the revenue of any $A_i \in agg(\hat{A})$ can be improved. □

## A.5 Additional Experiments

We now turn to a slightly larger scale experiment: $n = 3$, $m = 10$, bidders valuations uniformly distributed between $[0, 1]$. We again pick three scenarios where the bidders' participation probabilities are independent, thus the three scenarios will share the same set of participation profiles, which are $g_1 =< 1, 1, 1 >, g_1 =< 0, 1, 1 >, g_1 =< 1, 0, 1 >, g_1 =< 1, 1, 0 >, g_1 =< 1, 0, 0 >, g_1 =< 0, 1, 0 >, g_1 =< 0, 0, 1 >, g_8 =< 0, 0, 0 >$. However, in each of the three scenarios, the participation profiles will take place with different probability.

Table 4: Probability of participation profiles

| Scenario | $g_1$ | $g_2$ | $g_3$ | $g_4$ | $g_5$ | $g_6$ | $g_7$ | $g_8$ |
|---|---|---|---|---|---|---|---|---|
| 1 | 0.032 | 0.128 | 0.128 | 0.008 | 0.032 | 0.032 | 0.512 | 0.128 |
| 2 | 0.175 | 0.175 | 0.175 | 0.075 | 0.075 | 0.075 | 0.175 | 0.075 |
| 3 | 0.441 | 0.189 | 0.189 | 0.049 | 0.021 | 0.021 | 0.091 | 0.009 |

We perform the same experimental procedure as we did for the $2 \times 2$ experiment, except the allocation MLP, payment MLP, and misreport MLP are all expanded to have depth of 17 and width of 120. We also only loop the misreport module 100 times with no early stopping on each regret estimation step. The auction function optimization step is also looped for 600 times with learning rate of $5 \times 10^{-4}$ for the first 200 iterations, $5 \times 10^{-5}$ for the next 200 iterations, and $5 \times 10^{-6}$ for the last 200 iterations.

Table 5: Performance of three learned auctions in their corresponding scenario after training on the tweaked dataset

| Scenario | $rev'_{agg}$ | $rev^*_{agg}$ | $rgt'_{agg}$ | $(\sqrt{rev'_{agg}} - \sqrt{rgt'_{agg}})^2$ |
|---|---|---|---|---|
| 1 | 3.725 | $\approx 3.378$ | 0.139 | 2.426 |
| 2 | 4.285 | $\approx 4.183$ | 0.095 | 3.105 |
| 3 | 4.696 | $\approx 4.817$ | 0.093 | 3.47 |

Table 6: Performance of a neural network trained on untweaked dataset tested on the three scenarios

| Scenario | $rev'_{agg}$ | $rev^*_{agg}$ | $rgt'_{agg}$ | $(\sqrt{rev'_{agg}} - \sqrt{rgt'_{agg}})^2$ |
|---|---|---|---|---|
| untweaked | 5.172 | $\approx 5.55$ | 0.065 | 4.076 |
| 1 | 0.816 | $\approx 3.377$ | 0.014 | 0.612 |
| 2 | 1.918 | $\approx 4.183$ | 0.03 | 1.469 |
| 3 | 3.3 | $\approx 4.817$ | 0.047 | 2.56 |

Table 7: Benchmark performance of learned aggregated auction on individual auctions ($^*$ implies estimated value).

| | | Scenario 1 | Scenario 2 | Scenario 3 |
|---|---|---|---|---|
| | $rev'_1$ | 5.334 | 5.392 | 5.427 |
| $g_1 =< 1, 1, 1 >$ | $rev^*_1$ | $\approx 5.55$ | $\approx 5.55$ | $\approx 5.55$ |
| | $rgt'_1$ | 0.586 | 0.086 | 0.086 |
| | $rev'_2$ | 5.463 | 5.090 | 5.068 |
| $g_2 =< 0, 1, 1 >$ | $rev^*_2$ | $\approx 4.55$ | $\approx 4.55$ | $\approx 4.55$ |
| | $rgt'_2$ | 0.261 | 0.145 | 0.119 |
| | $rev'_3$ | 4.754 | 5.122 | 5.095 |
| $g_3 =< 1, 0, 1 >$ | $rev^*_3$ | $\approx 4.55$ | $\approx 4.55$ | $\approx 4.55$ |
| | $rgt'_3$ | 0.456 | 0.16 | 0.149 |
| | $rev'_4$ | 4.995 | 5.067 | 5.014 |
| $g_4 =< 1, 1, 0 >$ | $rev^*_4$ | $\approx 4.55$ | $\approx 4.55$ | $\approx 4.55$ |
| | $rgt'_4$ | 1.468 | 0.13 | 0.207 |
| | $rev'_5$ | 0.002 | 3.651 | 3.287 |
| $g_5 =< 1, 0, 0 >$ | $rev^*_5$ | $\approx 3.47$ | $\approx 3.47$ | $\approx 3.47$ |
| | $rgt'_5$ | $2 \times 10^{-9}$ | 0.116 | 0.163 |
| | $rev'_6$ | 4.428 | 3.517 | 3.625 |
| $g_6 =< 0, 1, 0 >$ | $rev^*_6$ | $\approx 3.47$ | $\approx 3.47$ | $\approx 3.47$ |
| | $rgt'_6$ | $5 \times 10^{-5}$ | 0.087 | 0.14 |
| | $rev'_7$ | 4.06 | 3.676 | $6 \times 10^{-4}$ |
| $g_7 =< 0, 0, 1 >$ | $rev^*_7$ | $\approx 3.47$ | $\approx 3.47$ | $\approx 3.47$ |
| | $rgt'_7$ | 0.103 | 0.131 | $1 \times 10^{-4}$ |

