# OpenReview forum: "Credible, Sealed-bid, Optimal Repeated Auctions With Differentiable Economics"
_ICLR.cc/2023/Conference — Submitted to ICLR 2023_

### Official Review · Reviewer_eKk8 · 2022-10-23

**Confidence:** 4
**Clarity, Quality, Novelty And Reproducibility:** See numbered items above
**Correctness:** 2
**Technical Novelty And Significance:** 1
**Empirical Novelty And Significance:** 1
**Recommendation:** 3

**Strength And Weaknesses:**

1.	You abstract the audit process, and essentially use an off-the-shelf auction learning algorithm. It feels like there isn’t much novelty left in your submission.
2.	I don’t think that your bid-revealing protocol resolves the seller credibility issue. An incredible seller could have a dummy buyer participating in the auction and submitting bids optimized to raise the price for the winning bidder. See e.g. https://arxiv.org/abs/2205.14758 on how to do it right.
3.	When considering repeated auctions, it is sub-optimal to use even the “optimal” single-shot auction many times. There is a rich literature on dynamic mechanisms that you should go read.
4.	It is not clear from your model how do bidders choose bids. Do they also do some kind of regret minimization? What happens when the auction uses a lottery?

**Typos:**

•	“Forty years of little analytical results” – what?! There are hundreds of papers on this. They don’t get fully optimal solutions for every case, but this is super well studied.

•	“Despite differentiable econ..” -> “Although…”

•	I can’t find the reference for Mor (2022)

•	“bids(i.e. “ -> “bids (i.e.~”

•	“the forgo the cost”

•	Algo 1 pseudocode: what is t not-greater-than T?! Did you mean \le?

•	“pay for together”

•	“and ran it” -> run

•	“the the weights”


**Summary Of The Paper:**

The authors consider the problem of a monopolist seller auctioning multiple items to many buyers, repeatedly over many time periods. They propose to learn the single-shot approximately IC mechanism using essentially off-the-shelf deep learning approaches (I guess this is why they submitted to ICLR?). Then they propose to solve the seller’s credibility issue by revisiting a random time period, revealing the bids and punishing the seller if they violated the auction terms.

**Summary Of The Review:**

See numbered items above

---

> ### Author Response · Authors · 2022-11-18
> **Response to Reviewer eKk8**
>
> We thank the reviewer for their comments and advice. We will fix the typos they pointed out in a revised version and the following is our response to their concerns.
>
> -----------------
>
> 1. Regarding potential dummy bidder breaking credibility:
>
> We appreciate this feedback, this wouldn't be a problem. The issue you are concerned with can certainly appear in the ascending auction, which is not sealed-bid and in which the winning bid is not only dependent on the valuation distribution of each bidder but also on the sequence of their actions (raise/exit). In this paper, we are considering sealed-bid auctions (e.g. 1st price, 2nd price), where the bidders together privately submit their bids to the auctioneer and the auctioneer determines who wins and how much they pay. The important distinction here is “sealed-bid”, which means no bidder knows the bids of any other bidder. It follows that no bidder can change other bidders’ bids with their own bids. Therefore, if the spy bidder were to place a strategic bid that raises the winning bid, they need to know the bids of other bidders and then determine some bid that raises the price for the winner according to the auction function (we provided an example in the introduction for 2nd price), then submit this bid. But this delay is impossible since the bidders have to make commitments together before submitting their bids to the auctioneer.
>
> Thank you for tagging https://arxiv.org/pdf/2205.14758.pdf, it is relevant as it also considers auction credibility, but its perspective is quite different from ours for the following reasons:
>
> (a) It is not considering sealed-bid auctions, which is a significant consideration in our paper and part of the trilemma in the original Akbarpour and Li paper (the previous paragraph provides more details on this).
>
> (b) It is considering single-item auctions while we are considering multi-item auctions
>
> (c) its goal is to find a specific auction, while ours is to determine a framework that works for learned auctions in general
>
> Nevertheless, it is relevant enough to be included in our related work section. We will update this in the revised version.
>
> 2. Regarding optimal repeated auctions:
>
> In our model, we assume the bidders are not online-learning agents, and we are not considering extensive-form game equilibria. We consider each round as independent of the others. We recognize that this is not made clear in the first submission, and we will strive to clarify it in a revised version.
>
> 3. Regarding how bidders choose bids:
>
> We shouldn’t need to worry about how bidders choose bids, because the auction learning procedure will find an auction that is (almost) incentive compatible, so it is safe to assume that bidders bid truthfully. Detail about this procedure can be found in sections 5.3 and 5.4. Our neural network architecture allows the auction to take on the form of a lottery, so if the auction happens to be a lottery, (almost) incentive compatibility is still achieved. We do note that current methods for learning optimal auctions only find an approximate version of incentive compatibility, more on its robustness: https://proceedings.neurips.cc//paper/2020/file/3465ab6e0c21086020e382f09a482ced-Paper.pdf.

---

### Official Review · Reviewer_cqog · 2022-10-24

**Confidence:** 4
**Correctness:** 4
**Technical Novelty And Significance:** 3
**Empirical Novelty And Significance:** 3
**Recommendation:** 8

**Clarity, Quality, Novelty And Reproducibility:**

The paper is well-organized and easy to follow. The study on credibility for neural auction design, especially combining cryptographic tools, is novel. The experiments are clearly described with high reproducibility.

**Strength And Weaknesses:**

Strength:
1. The issue of credibility is important for the field of differentiable economics.
2. The idea of introducing cryptographic tools, although straightforward, is quite convicing. The combination of verification and repeated auctions is properly designed, with clear theoretical analysis for credibility guarentee and corresponding revenue.

Weaknesses:
1. The assumption of stationary participation may be kind of strong. Nevertheless, I am not sure whether there has been related works on this issue.

**Summary Of The Paper:**

The paper studies how to combine cryptographic tools with neural auction design to provide credible optimal (repeated) auctions. The idea is to adding a verification step in repeated auctions to check the credibility of the auctioneer, while with properly design penalty, only excuting verification for a limited times is enough to prevent auctioneer's cheating. The repeated auction desgin through deep learning is transfered from previous work for single round auction, with an aggregation under bidders' stationary participantion. The approximate revenue maximization is supported by experiments.

**Summary Of The Review:**

The paper consider an important issue on credibiliy in differentiable auctions. The designed protocol is convincing with both theoretical analysis and experiments. Overall, I would recommend acception.

---

> ### Author Response · Authors · 2022-11-18
> **Response to Reviewer cqog**
>
> We thank the reviewer for their insights. We agree that the stationary participation assumption may be strong. We would also like to point out that previous works (Duetting et al., Rahme et al., Peri et al., etc) in the differentiable economics assume fixed bidders (they are only considering one round of auction, so this makes sense), so we have actually weakened this assumption on bidder participation.

---

> > ### Comment · Reviewer_cqog · 2022-11-18
> > **Response**
> >
> > Regarding the participatin issue, one suggestion is to consider using the contexts to distinguish the bidders participating in each round. (ref: A Context-Integrated Transformer-Based Neural Network for Auction Design, ICML 2022)

---

> > > ### Author Response · Authors · 2022-11-25
> > > **Thank you for your suggestion**
> > >
> > > Thank you for your kind suggestion. The paper you tagged is a high-quality paper that studies bidder participation more nuancedly than we did. We would revise our paper to add a discussion of this work if our paper were to be accepted.

---

### Official Review · Reviewer_zy43 · 2022-10-25

**Confidence:** 2
**Correctness:** 4
**Technical Novelty And Significance:** 3
**Empirical Novelty And Significance:** 3
**Recommendation:** 8

**Clarity, Quality, Novelty And Reproducibility:**

The paper is well-written and easy to read. The claims made in the paper are sufficiently demonstrated through experiments. The contributions are quite novel as well.

**Strength And Weaknesses:**

**Strengths:** The proposed framework is quite novel. The authors also perform sufficient experiments to illustrate the efficiency of the training procedure. This direction of research of also quite relevant and is of significant importance especially given that a lot of tech companies generate revenue through auctions.

**Weakness:** The approach is not quite scalable, although I think this is a good first step.


**Summary Of The Paper:**

It is known that no credible, sealed-bid, and incentive compatible exists, assuming no communication between the bidders. In this work, the authors relax this assumption and propose a framework to run efficient, credible, and revenue-optimal repeated auctions with cryptographic tools. Their main contributions are as follows:
1. The authors show how an auction can be made credible with just a verification scheme and a penalty. This verification scheme could simply be to bring in a trusted third party for auditing. They also show how to set a penalty that is independent of the auctioneer being untruthful.
2. The authors also propose a stationary bidder participation model and a method for training revenue-maximizing auctions in this context. They demonstrate the efficacy of this approach through a couple of experiments.

**Summary Of The Review:**

Overall this is a good paper. The use of differential economics for running credible, incentive-compatible, and revenue-maximizing auctions is a new and an exciting direction to pursue.

---

> ### Author Response · Authors · 2022-11-18
> **Response to Reviewer zy43**
>
> We thank the reviewer for their comments and advice. We agree with the reviewer's point about the approach not being scalable.

---

### Official Review · Reviewer_53Tu · 2022-10-26

**Confidence:** 3
**Correctness:** 4
**Technical Novelty And Significance:** 2
**Empirical Novelty And Significance:** 2
**Recommendation:** 3

**Clarity, Quality, Novelty And Reproducibility:**

The paper studies an interesting setup, but both the theoretical and empirical contributions are somewhat limited. The paper is a bit difficult to follow since there appear to be two separate and somewhat disjointed contributions. Moreover, the notation a bit awkward, with the superscripts appearing to the left of the variables.

**Strength And Weaknesses:**

Strengths:
- The idea of using cryptographic verification schemes to guarantee commitment of the auctioneer is interesting. It provides a different approach to ensure credibility than is typically studied in market design (it is obtained by relaxing the assumption of no bidder communication).

Weaknesses:
- The theoretical contribution is limited. The proof of credibility is straightforward, since the penalty is set so high that it guarantees that the auctioneer would rather stick to the proposed mechanism than deviate. It may be more interesting to investigate tradeoffs between approximate credibility and penalties.
- The empirical contribution—designing revenue-maximizing auctions in the repeated auction setup—seems incremental relative to existing work in the automated auction design literature (e.g. Duetting et al., 2019, Rahme et al., 2020).
- The two contributions in this paper—proposing a scheme to guarantee credibility and training revenue-maximizing auctions in the repeated auction setup when bidders can change between rounds—are somewhat disjointed. It is perhaps confusing to combine these investigations into a single paper.


**Summary Of The Paper:**

The paper examines how to design credible auctions (i.e. auctions where the auctioneer is not incentivized to deviate from their proposed mechanism), with a focus on repeated auctions. The setup is that the auctioneer proposes an allocation rule and a payment function at the beginning of the game, and then they run a multi-item auction at each time step. The paper proposes using a cryptographic verification scheme and penalties at randomly chosen set of time steps to guarantee credibility.

The paper then considers the design of revenue-maximizing auctions in the repeated auction setup in the case where not all bidders necessarily participate in the auction at every time step (they change according to a stationary probability distribution). They propose a method for training revenue-maximizing auctions in this model and evaluate the method in simulations on small auction setups.

**Summary Of The Review:**

The paper studies an interesting setup of designing credible auctions in the repeated auctions setup. However, both the theoretical and empirical contributions are somewhat incremental relative to prior work.

---

> ### Author Response · Authors · 2022-11-18
> **Response to Reviewer 53Tu**
>
> We appreciate the reviewer for their comments and advice. The following is our response to their concerns.
>
> W1: Theoretically, the high penalty does not harm any desirable properties of the auction (credibility, incentive compatibility, sealed-bid, etc.). In practice, a high penalty is a number that appears on the contract, which the auctioneer would be able to avoid if they were truthful. Therefore we do not think it is necessary to investigate the tradeoff between the penalty and credibility.
>
> W2: We agree that the contributions to automated mechanism design are limited. In this paper, we are primarily concerned with addressing auction credibility, and not so much with traditional concerns in automated mechanism design, such as revenue, incentive compatibility, robustness, etc. Our contribution to automated mechanism design is related to the stationary participation model, in which we propose to extend the work of Duetting et al., and Rahme et al., to a larger class of auctions. The proof may appear straightforward, but the results are quite powerful and haven’t been done before. Furthermore, the stationary participation model is especially relevant to our solution to credibility, as it also extends our credibility solution to a larger class of auctions. We admit that this connection is not made clear in the first submission, more on this in the response below.
>
> W3: We appreciate this feedback and agree that the connection is somewhat unclear. The relevance of the stationary participation model appears naturally if we were to actually apply the protocol developed in section 4. Previous works in automated mechanism design (Duetting et al., Rahme et al., etc.) are mostly concerned with stage auctions, so it is natural to assume that the bidders are fixed. In our work, the efficiency of the credible auction protocol is dependent on the repetition of the auctions over many rounds, so it is unrealistic to assume that the bidders are fixed from start to end. Thus we proposed the stationary participation model to determine what the optimal auction should look like when bidder participation is uncertain, we then find that such an auction will look like an “aggregated auction” (defined in the paper), and can be approximated using a neural network. In theory, the protocol we proposed in section 4 works with any neural network-encoded auction, so this means we have found a way to run credible and optimal auctions when bidders participate with uncertainty, a much more realistic assumption. We will revise the paper to make this connection clear.

---

### Decision · Program_Chairs · 2023-01-20

**Decision:**

Reject

**Justification For Why Not Higher Score:**

Two experts have multiple justified requests for improvement.

**Justification For Why Not Lower Score:**

Sensible paper, important topic, authors have expertise.

**Metareview: Summary, Strengths And Weaknesses:**

(a) Provides a cryptographic mechanism that incentivizes an auctioneer not to cheat in repeated auctions. Also shows how to use gradient descent to design a repeated auction when bidders may be different in different rounds.

(b) Highly relevant to a $100B industry. Likely correct.

(c) Reviewers are split and two experts out of four have many criticisms.

Overall, the authors can earn from the reviews and resubmit. They may want to take the advice to divide into two papers.

Comment from the AC: The authors say "Overall, we don't expect bidders dropping out to be an issue unless the application setting is radically different from the model we studied." Well, this is a common mechanism design on Wall Street, called "last look." In general, the authors should pay attention not just to ad auctions as the application area, but also to financial trading.



**Summary Of Ac-Reviewer Meeting:**

No meeting.